# Amyloid pathology disrupts gliotransmitter release in astrocytes

**Anup Gopalakrishna Pillai**, **Suhita Nadkarni** *

Indian Institute of Science Education and Research Pune, Pune, India

* suhita@iiserpune.ac.in

**Data Availability Statement:** All the codes of our model and simulations will be made publicly available at the GitHub repository: https://github.com/anupgp/astron.

**Funding:** AGP was supported by a Postdoctoral Fellowship from the Indian Institute of Science Education and Research Pune (IISER-P/Ext/PDRF/

## Abstract

Accumulation of amyloid-beta (Aβ) is associated with synaptic dysfunction and destabilization of astrocytic calcium homeostasis. A growing body of evidence support astrocytes as active modulators of synaptic transmission via calcium-mediated gliotransmission. However, the details of mechanisms linking Aβ signaling, astrocytic calcium dynamics, and gliotransmission are not known. We developed a biophysical model that describes calcium signaling and the ensuing gliotransmitter release from a single astrocytic process when stimulated by glutamate release from hippocampal neurons. The model accurately captures the temporal dynamics of microdomain calcium signaling and glutamate release via both kiss-and-run and full-fusion exocytosis. We investigate the roles of two crucial calcium regulating machineries affected by Aβ: plasma-membrane calcium pumps (PMCA) and metabotropic glutamate receptors (mGluRs). When we implemented these Aβ-affected molecular changes in our astrocyte model, it led to an increase in the rate and synchrony of calcium events. Our model also reproduces several previous findings of Aβ associated aberrant calcium activity, such as increased intracellular calcium level and increased spontaneous calcium activity, and synchronous calcium events. The study establishes a causal link between previous observations of hyperactive astrocytes in Alzheimer's disease (AD) and Aβ-induced modifications in mGluR and PMCA functions. Analogous to neurotransmitter release, gliotransmitter exocytosis closely tracks calcium changes in astrocyte processes, thereby guaranteeing tight control of synaptic signaling by astrocytes. However, the downstream effects of AD-related calcium changes in astrocytes on gliotransmitter release are not known. Our results show that enhanced rate of exocytosis resulting from modified calcium signaling in astrocytes leads to a rapid depletion of docked vesicles that disrupts the crucial temporal correspondence between a calcium event and vesicular release. We propose that the loss of temporal correspondence between calcium events and gliotransmission in astrocytes pathologically alters astrocytic modulation of synaptic transmission in the presence of Aβ accumulation.

AP-20145047/10/2016), Google Cloud research credits (EDU Credit kcapozzi 206784490) and Wellcome Trust/DBT India. SN was funded by Wellcome Trust/DBT India Alliance (IA/I/12/1/500529) and by the Indian Institute of Science Education and Research Pune.The funders had no role in study design, data collection and analysis, decision to publish, or preparation of the manuscript.

**Competing interests:** The authors have declared that no competing interests exist.

## Author summary

Signaling by astrocytes is critical to information processing at synapses, and its aberration plays a central role in neurological diseases, especially Alzheimer's disease (AD). A complete characterization of calcium signaling and the resulting pattern of gliotransmitter release from fine astrocytic processes are not accessible to current experimental tools. We developed a biophysical model that can quantitatively describe signaling by astrocytes in response to a wide range of synaptic activity. We show that AD-related molecular alterations disrupt the concurrence of calcium and gliotransmitter release events, a characterizing feature that enables astrocytes to influence synaptic signaling.

## Introduction

Astrocytes, the most abundant glial cells, are now recognized for their role in maintaining normal brain functioning [1,2]. Apart from providing structural and energy support to neurons, these densely connected cells send ramified processes to neighboring synapses and form non-overlapping synaptic islands [3,4]. A plethora of receptors are found on the astrocytic membrane; amongst them, metabotropic glutamate receptors (mGluRs) are specifically juxtaposed to the endoplasmic reticulum (ER) tubules and are expressed in high densities on compartments adjacent to synapses [5–7]. This remarkable configuration and the presence of inositol 1,4,5-trisphosphate receptor (IP$_3$Rs) clusters on the ER allow fast and high amplitude calcium transients in the ramified processes of the astrocytes [8,9]. Along with these molecular components for Ca$^{2+}$ signaling and their placement [10,11], astrocytes are also equipped with an elaborate apparatus for fast Ca$^{2+}$-dependent release of gliotransmitters, which is comparable to neurons [12,13]. The link between calcium signaling and gliotransmission from astrocytic microdomains was established by a study that quantified Ca$^{2+}$ transients using near-field imaging of astrocytic terminals loaded with a low-affinity calcium indicator in combination with a pH-sensitive vesicular glutamate transporter [5]. This technique allowed them to track subcellular Ca$^{2+}$ events and the corresponding gliotransmitter release events simultaneously with good temporal resolution and brought to light several novel features of gliotransmission at subcellular compartments near synapses. First, the application of a short pulse of mGluR agonist evoked fast and stochastic Ca$^{2+}$ events whose time course perfectly matched the ensuing gliotransmitter release events. The study also revealed tight spatial and temporal correlations between Ca$^{2+}$ and exocytic fusion events. These results strongly indicate that gliotransmitter release events are coupled to microdomain Ca$^{2+}$ events in individual astrocytic compartments. The study further reports the presence of two distinct classes of vesicle populations that differ in their fusion modes (kiss-and-run versus full-fusion) and in the timescales of endocytosis and reacidification. Several synaptotagmin (*Syt*) isoforms (calcium sensors for vesicle release) have been reported in astrocytes; amongst the subtypes, *Syt4* and *Syt7* are specifically associated with fast synchronous kiss-and-run and slow asynchronous full-fusion releases, respectively [14–17]. Several other studies also provide compelling evidence on the existence of dual release pathways and the ability of astrocytes to modulate synaptic strength in neuronal circuits [18–21].

The intimate spatial association between astrocytic compartments and synaptic junctions, together with the presence of a fast Ca$^{2+}$ signaling machinery and multiple pathways for gliotransmitter release, provide the necessary framework for astrocytes to engage in bidirectional communication with neurons [22,23]. However, this pathway, an intrinsic component of communication by astrocytes, is heavily compromised under calcium overload, leading to

excitotoxicity, loss of spines, and network connectivity [24,25]. Of relevance to the present study is the correlation between Aβ plaques seen in AD and the disruption of glutamate-mediated excitatory transmission [26]. Multiple lines of evidence point to a link between Aβ toxicity and calcium dysregulation in astrocytes [27]. Not only is Aβ a direct modulator of astrocytic $Ca^{2+}$ [28], but it also elevates resting $Ca^{2+}$ levels and synchronous $Ca^{2+}$ events [29].

Even though several studies report on abnormal $Ca^{2+}$ regulation in astrocytes exposed to Aβ, not much is known about the chronology of events and molecular mechanisms that disrupt the calcium signaling [30]. Nevertheless, both in vitro and post-mortem studies in AD brains highlight the role of two important $Ca^{2+}$ regulating pathways, namely mGluRs [31,32] and PMCAs [33,34]. It has also been suggested that a significant part of the excess glutamate observed in AD brains has an astrocytic origin and is a consequence of Aβ signaling [35]. Again, the underlying biophysical links are not clear. These studies underscore that cellular degeneration in AD is orchestrated by a chain of signaling cascades extending all the way from Aβ to alterations in $Ca^{2+}$ signaling and synaptic signaling by astrocytes [36]. Understanding the astrocytic mechanisms that are derailed under the influence Aβ is, therefore, crucial for advancing our knowledge of the pathogenesis of synaptic dysfunction that underlies the cognitive decline in AD [37].

Synergistic crosstalk between theory and experiments has led to some of the most profound insights in neuroscience and, more specifically into neurotransmitter release machinery and organization [38–40]. However, an equivalent biophysical modeling framework for calcium-dependent vesicle release and recycling does not exist, despite extensive knowledge on the calcium-binding kinetics of synaptotagmins [41], their different modes of vesicle release [18,42] and localization and lastly, calcium signaling [5] in the ramified processes that envelop synapses. Previous computational models have focused on global $Ca^{2+}$ signals in astrocytes [43–45] and proposed phenomenological models to capture feedback to neurons via gliotransmission at a tripartite synapse [46–48]. Recently, studies have also shed light on the microscale mechanisms at fine astrocytic processes [49–52]. These models have made several valuable predictions on the contribution of astrocytes to brain function. The present study builds on this literature to fill the conspicuous gap (biophysical model of vesicle release and recycling) in the models and extend the existing computational modeling framework [53,54].

The primary route for $Ca^{2+}$ excitability in astrocytes in our model is through the activation of mGluRs that are expressed in high levels on astrocytic compartments around synapses [2,55,56]. A limited number of IP$_3$Rs on the ER generate fast but noisy $Ca^{2+}$ transients in the astrocytic process [5]. We implement stochastic opening and closing of a small cluster of IP$_3$Rs consistent with experimental studies. Our model also accounts for the high luminal $Ca^{2+}$ concentration and ER/cytosol volume ratio in the process compared to the cell body [5,57,58]. The elaborate machinery available to astrocytes for fast $Ca^{2+}$-dependent release of gliotransmitters has been reported extensively [12,13]. We model molecular details as well as the timescales of calcium-binding for *Syt4* and *Syt7* isoforms of Synaptotagmins (*Syt*) that trigger fast synchronous kiss-and-run and slow asynchronous full-fusion releases, respectively, in astrocytes [14–17]. Our biophysical model accurately chronicles experimentally observed temporal dynamics of high amplitude and fast $Ca^{2+}$ events when stimulated by mGluR agonist and exo-endocytosis steps for $Ca^{2+}$-mediated gliotransmitter release [1,2,5,11,13,58–60]. We use this validated biophysical model for calcium signaling and the resulting gliotransmitter release by an astrocyte at a tripartite synapse to examine the functional interplay between two key $Ca^{2+}$ regulating mechanisms that are altered by Aβ. Furthermore, we predict the consequences of altered $Ca^{2+}$ signaling on gliotransmission at astrocytic microdomains for a wide range of stimuli and describe the underlying mechanism.

## Methods

The governing equations for $Ca^{2+}$ and $IP_3R$ signaling components of the model (Fig 1A) are presented in S1 Appendix. Our choice of parameter values (S1 Table) are based on published literature and were adjusted, if necessary, to reproduce the fast kinetics of mGluR-dependent $Ca^{2+}$ transients reported by previous studies [5,10,61].

### Glutamate stimulation

Glutamate evoked $Ca^{2+}$ events, and gliotransmission at a single astrocytic compartment was examined by stimulating with either steady-state pulses (2s & 30s) or by a time varying profile

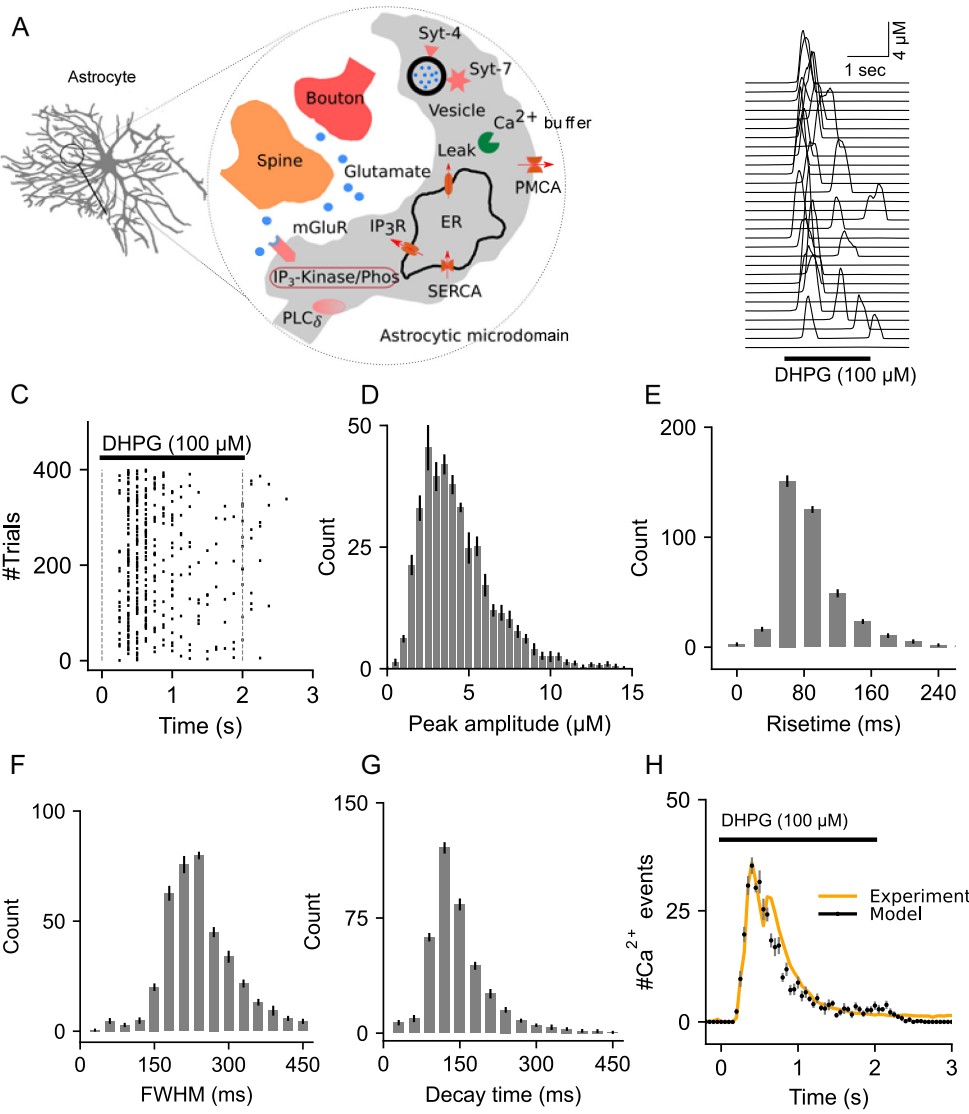

**Fig 1. Description and characterization of $Ca^{2+}$ signaling in the astrocytic compartment.** (A) Schematic of the astrocytic compartment along with all the molecular components. (B) Representative $Ca^{2+}$ events from an astrocytic compartment stimulated with DHPG (100 μM, 2 sec). (C) Raster plot of $Ca^{2+}$ events. (D) Distribution of $Ca^{2+}$ event peak amplitudes from 400 trials. (E) Distribution of $Ca^{2+}$ event rise times. (F) Histogram of full width at half maximum (FWHM). (G) Decay time histogram. (H) Temporal distributions of $Ca^{2+}$ events from the model and experimental data (Marchaland et al., 2008a). Error bars indicate the standard error of the mean computed from 6 independent datasets.

of glutamate concentration (see below) as observed at the perisynaptic region close to astro-cytic compartments, which was kept the same for both control and Aβ conditions [62–64]. Fol-lowing each presynaptic spike, a pulse of glutamate was released, whose time course was implemented using a simple first-order reaction equation that captured both the sharp peak (~200μM) and decay (6.25ms) of glutamate concentration (S1 Appendix and S1 Table and S1A Fig). The effective glutamate concentration ([Glu](t)) was modelled using the below equa-tion whose parameters were adjusted to consider changes in the glutamate concentration due to mechanisms such as receptor binding and reuptake.

$$[\text{Glu}](\text{t}) = Glu_{max}\delta(\text{t}-\text{t}_r) + [\text{Glu}]\exp(k_{Glu}\text{t}), \ \text{t}_r = \text{glutamate release time} \quad\quad 1$$

## IP$_3$ dynamics

While the major IP$_3$ generation pathway was through the activation of mGluRs by glutamate spillover from adjacent presynaptic terminals, IP$_3$ was also generated in a Ca$^{2+}$-dependent manner through PLC$_\delta$ activity. The model also included two IP$_3$ degradation pathways, namely, inositol polyphosphate 5-phosphatase and IP$_3$ 3-kinase [65–67] that are described in S1 Appendix, and the parameters are in S1 Table.

## Modeling Ca$^{2+}$-mediated gliotransmission

Both fast kiss-and-run confined releases and slow-spreading full-fusions events have been reported by studies that imaged the release of fluorescent labeled vesicles from astrocytic pro-cesses [18,20]. Among the important molecular components that govern Ca$^{2+}$-mediated release, two synaptotagmins (*Syts*), namely *Syt4* and *Syt7*, have been associated with rapid kiss-and-run and slow full-fusion exocytosis, respectively, have been found in astrocytes [14,15,17,68]. Moreover, *Syt4*, which is known to specifically promote kiss-and-run exocytosis [69], has a single low-affinity ($k_d \sim 22$ μM) domain that binds Ca$^{2+}$ ions with fast forward cal-cium-binding rates, making it a detector of fast and high amplitude Ca$^{2+}$ events. Whereas, *Syt7*, has five high affinity ($k_d \sim 15$ μM) binding sites that bind Ca$^{2+}$ at a much slower rate [41] and is thought to mediate full-fusion release when activated by slow and low amplitude Ca$^2$$^+$events. The involvement of *Syt7* in slow full-fusion exocytosis is further in agreement with data from an experiment that measured changes in cell capacitance following slow full-fusion release events [70]. Taking these studies into account, we modeled Ca$^{2+}$-dependent gliotrans-mitter release using detailed kinetics schemes of two distinct synaptotagmins (*Syt4* and *Syt7*). We also incorporated detailed kinetics of endocytosis, vesicle recycling, and vesicle docking as reported experimentally [5,71,72] (Fig 2B and S1 Table). Moreover, as suggested by one study [73], we considered two pools of vesicles (docked and mobile) that are distinctly targeted by *Syt4* & *Syt7*. Our model assumes newly recycled vesicles to remain as mobile vesicles until they are transported to the docking site. Accordingly, we introduced a docking rate that controls the transfer of vesicles from the mobile pool to the docked vesicle pool (Fig 2B). Modeling the distinct Ca$^{2+}$ binding sites and reaction rates of the two synaptotagmins (*Syt4* & *Syt7*) and their target vesicle pools was crucial for reproducing the time courses of kiss-and-run and full-fusion releases as observed experimentally (Fig 2B and S1 Table). In summary, high-affinity binding sites in *Syt7* promote slow full-fusion releases when activated by low amplitude Ca$^{2+}$ elevations, while fast low-affinity binding sites in *Syt4* promote rapid kiss-and-run release of docked vesicles in the presence of high amplitude Ca$^{2+}$ rises. Each of these pathways also fol-lows different rates of endocytosis and recycling and has been modeled accordingly [74].

A typical chain of events starts with a presynaptic release followed by: (1) rise in extracellu-lar glutamate levels, (2) activation of mGluRs on the astrocytic compartment, (3) increase in intracellular IP$_3$, and (4) stochastic opening of IP$_3$Rs on the ER membrane to generate fast

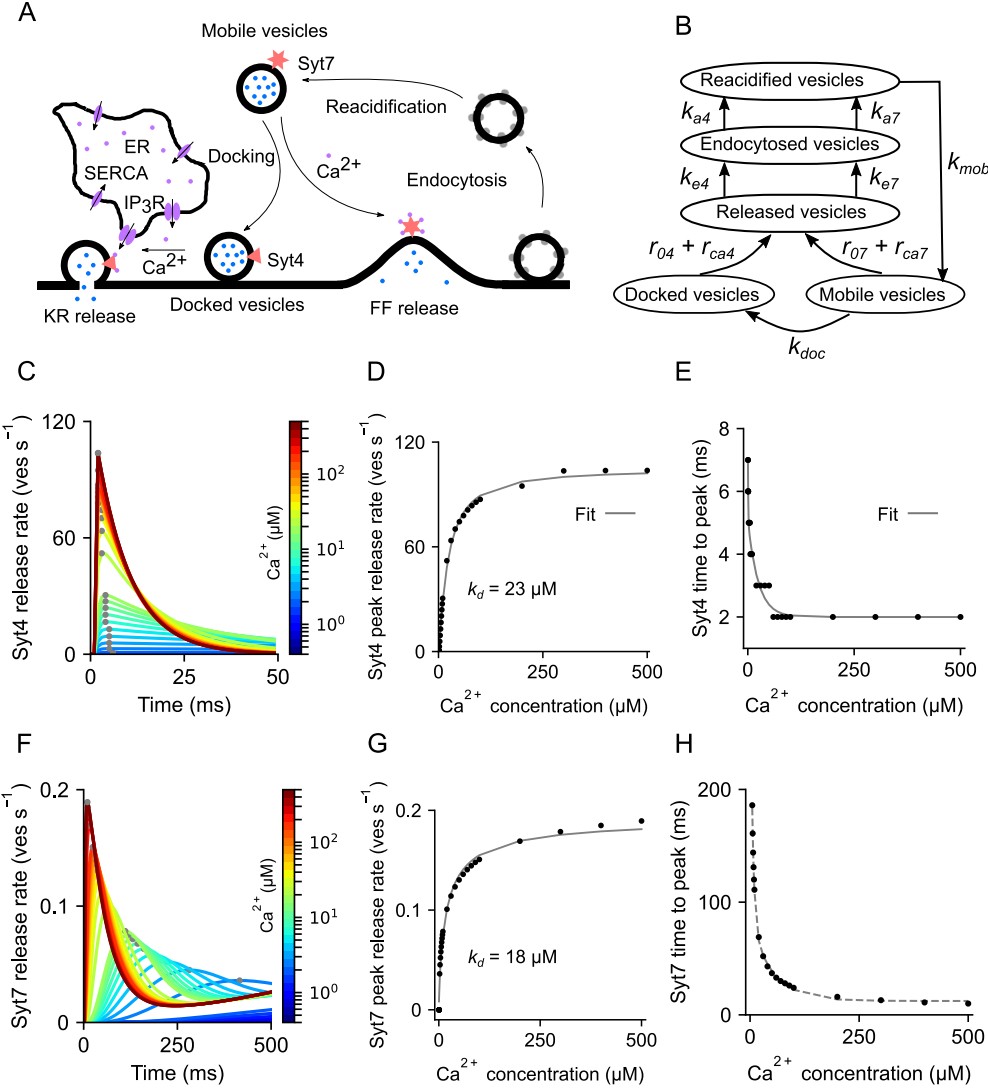

**Fig 2. Description and characterization of the Ca²⁺-dependent gliotransmitter release machinery.** (A) Schematic of the gliotransmitter release machinery. (B) Pathways of release, endocytosis, and recycling of docked and mobile vesicles. (C) Vesicular release rate through Synaptotagmin 4 (*Syt4*) in response to a range of steady state Ca²⁺ concentrations. (D) Peak release rates of *Syt4* are fitted to a Hill function to estimate the dissociation constant ($k_d$). (E) Time to peak release rate of *Syt4* decays exponentially with Ca²⁺ concentration. (F) Release rates of Synaptotagmin 7 (*Syt7*) in response to a range of steady state Ca²⁺ concentrations. (G) Peak release rates of *Syt7*. (H) Time to peak release rate of *Syt7*.

Ca²⁺ transients (S1 Fig). These Ca²⁺ events have low peak amplitudes (5–20 nM) when stimulated with presynaptic glutamate release at low frequencies. However, IP₃ concentration quickly builds up when the stimulation frequency is increased, leading to high amplitude Ca²⁺ events (5–10 μM) that can trigger gliotransmission. A release event gets triggered (either via *Syt4* or *Syt7)* every time the release probability, computed by multiplying the instantaneous release rate and the integration time step (50 μs), is greater than a uniform random number over the open interval (0,1). The Ca²⁺-dependence of the release machinery was subsequently characterized by clamping intracellular Ca²⁺ level and fitting the peak release rate to the below

equation with $V_{max}$, $k_d$, and $n$ as free parameters [75].

$$\text{Release rate } ([Ca_{cyt}]) = V_{max} \text{ Hill } ([Ca_{cyt}]^n, k_d) \qquad 2$$

## Quantifying the percentage of calcium response

The percentage of $Ca^{2+}$ responses ($r$) from different stimulation regimes was quantified by normalizing the area under the curve using the equation described below, where $t_0$ and $t_{stim}$ refer to stimulation onset and duration, respectively.

$$r = \frac{\int_{t_0}^{t_0 + t_{stim}} [Ca^{2+}]_{cyt}(t)dt}{\int_{t_0 - t_{stim}}^{t_0} [Ca^{2+}]_{cyt}(t)dt} \qquad 3$$

## Detection and measurement of calcium events

The full width at half maximum (FWHM) of $Ca^{2+}$ events whose peak amplitude crossed the threshold value (300 nM) was identified using the 'find_peaks' and 'peak_widths' functions available from SciPy-1.7.3. The choice of the threshold for identifying $Ca^{2+}$ events was based on a previous report on the physiological minimum to evoke glutamate release from astrocytes [76]. Rise time was measured as the time between 20% and 80% of the signal peak, and decay time was measured as the time between peak and 36.8% of the peak amplitude.

## Synchrony measure

Synchrony of discrete $Ca^{2+}$ and release events evoked by glutamate application at different rates was computed using the Pinsky-Rinzel method [77,78]. The data set comprised of 1000 independent trials ($j$) each for stimulation frequencies ranging from 0.4 to 100 Hz. Firstly, (1) a vector of inter-event intervals (k) was computed, with respect to each event, for all the events in the event matrix (events x trials). (2) The vector of inter-event intervals was normalized by the maximum interval to obtain the phase vector ($\phi$). (3) The phase vector was projected to a complex plane, and synchrony was computed from the (4) variance as given by the equations below. The above steps (1–4) were repeated by choosing another event as the reference event and the values were averaged. Synchrony ($s$) was computed for each stimulation frequency and inter-event interval pair, where a value of 1 corresponds to perfect alignment of event times across all the trials and, 0 means that the events are completely random.

$$Z(\phi_k) = e^{2\phi_k(j,m)\pi i} \qquad 4$$

$$s = \sqrt{1 - Var(Z(\phi_K))} \qquad 5$$

## Computation of cross-correlation between $Ca^{2+}$ and release events

Cross-correlations within a fixed time window (100ms) were computed from the diagonal elements of the normalized joint peristimulus time histogram of $Ca^{2+}$ and release event times [79,80]. Peak cross-correlations were extracted from the signal duration (11s) starting from the time of glutamate stimulation at frequencies ranging from 0.4 to 100 Hz. The standard error of the mean was estimated by sampling with replacement from a set of 1000 simulation trials.

## Numerical simulation and code

Simulation codes written in C++ were compiled and computed on the high-performance computing cluster (1500 nodes) at the Indian Institute of Science Education and Research Pune. Part of the simulations and data analysis were carried out on a desktop computer running

Linux kernel 5.16.11 with custom Python scripts using Python-3.10.2. State variables were updated at fixed time steps (50μs) using the Euler–Maruyama method for integrating stochastic differential equations. Standard errors in the plots were computed from independent sets of simulation runs or by sampling with replacement (bootstrapping) from a large set. Students t test and ANOVA were used to determine the statistical significance of the difference between means; $P < 0.05$ was considered statistically significant. $P < 0.05$, $0.01$ and $0.001$ are indicated in the figure by single, double and triple asterisks, respectively. Averaged values are reported as mean ± SEM. All the simulation and data analysis codes are available from the author's GitHub page (https://github.com/anupgp/astron).

## Results

### A single-compartment model for astrocytic Ca$^{2+}$ signaling and gliotransmission

We first describe local calcium dynamics in an astrocytic microdomain that envelops a typical hippocampal CA1 synapse (Fig 1A). Distinct molecular components on the plasma and ER membranes, Ca$^{2+}$ buffers in the cytoplasm and ER collectively regulate the fast and slow Ca$^{2+}$ events. Specifically, the stochastic opening of IP$_3$Rs on the ER membrane generates rapid Ca$^{2+}$ events whose rise time and decay kinetics are modulated by PMCA pumps on the plasma membrane, Ca$^{2+}$ buffers, and SERCAs.

A single pulse of DHPG (100 μM, 2 secs), a highly specific group 1 mGluR agonist, generates sharp and high amplitude Ca$^{2+}$ events during the stimulus duration (Fig 1B). Stochastic Ca$^{2+}$ events from independent trials are described in the raster plot in Fig 1C. The distributions of peak amplitude, rise time, FWHM, and decay time, are in good agreement with previous experimental observations (Fig 1D–1G) [11,61]. We next examined the temporal characteristics of these events from a set of 400 independent trials to validate our model findings with a previous study by Marchaland et al. that examined Ca$^{2+}$ signals from individual microdomains [6]. Our choice of the number of independent trials for comparing our results is based on a rough estimate of the number of microdomains included in the experimental data (S2 Appendix). The close overlap between the Ca$^{2+}$event histogram of the model and the experimental data (Fig 1H), and the good agreement with experimental data on Ca$^{2+}$ events characteristics described above, validates our model for a single astrocytic compartment.

We next describe the biophysical model for Ca$^{2+}$-mediated gliotransmitter release in a single astrocyte compartment, drawing on existing insights on Ca$^{2+}$-sensors (synaptotagmins), vesicle distribution, and recycling times (Fig 2A and S3 Appendix). The model incorporates two separate Ca$^{2+}$ sensors, namely *Syt4* and *Syt7*, each with distinct Ca$^{2+}$ binding affinities and forward reaction rates to mediate fast kiss-and-run and slow full-fusion releases, respectively. Analogous to vesicle recycling at the presynaptic terminal, following a release, vesicles are endocytosed, reacidified, and added to the mobile vesicle pool before they become available in the docked vesicle pool (Fig 2A and 2B). A detailed description of the release machinery is included in the Methods section (*Modeling Ca$^{2+}$-mediated gliotransmission*).

We first characterize the temporal dynamics of the two synaptotagmins that differed in their (1) binding affinities, (2) forward reaction rates, and (3) target vesicle pools by characterizing their release rates at various steady-state intracellular Ca$^{2+}$ levels. The *Syt4* activation led to large release rates that rapidly decayed in tens of milliseconds. In comparison, the peak release rate of *Syt7* is much lower and takes hundreds of milliseconds to get back to baseline (Fig 2C and 2F). We further quantify their decay kinetics and estimate equilibrium constants ($k_d$) from fits to double exponentials (Fig 2E and 2H) and Hill equations (Fig 2D and 2G), respectively. Our estimates of $k_d$ for *Syt7* agree with a previous study [59]. Our results also

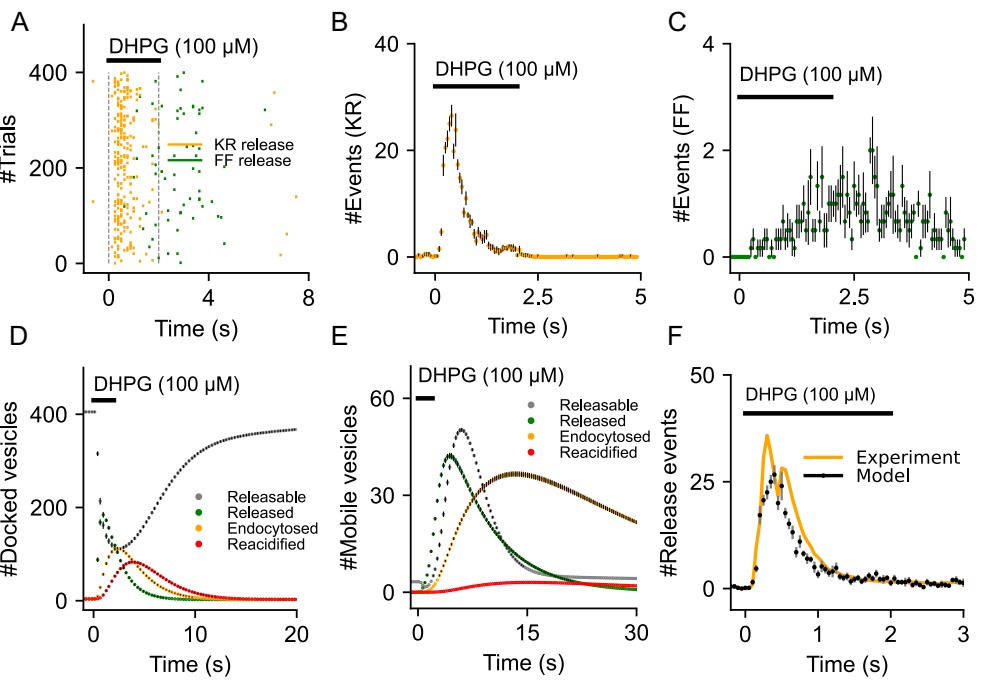

**Fig 3. Validation of the gliotransmitter release model.** (A) Raster plot of kiss-and-run and full fusion releases from an astrocytic compartment stimulated with DHPG (100 μM, 2 sec). (B) Distribution of kiss-an-run release times shows the rapid rise in release events that decay exponentially within the stimulus duration. (C) Temporal distribution indicates the comparatively slow and low number of full-fusion events that extends beyond the stimulus duration. (D) Time courses indicate the dynamics of docked vesicle release, endocytosis and reacidification. (E) Time courses indicate the dynamics of mobile vesicle release, endocytosis and reacidification. (F) Temporal distribution of total release events is in good agreement with experimental data from Marchaland et al. (Marchaland et al., 2008a). Error bars indicate the standard error of the mean computed from 6 independent sets of simulation trials.

showcase substantial differences between *Syt4* and *Syt7* in their (1) binding affinities, (2) rise times, and (3) decay kinetics.

We next examine the temporal features of gliotransmission when activated by a short pulse of DHPG (100 μM, 2 secs) (Fig 3A). Here, the gliotransmitter release event is probabilistically determined by the release rate of the two synaptotagmins, which in turn is influenced by both the amplitude and decay kinetics of individual $Ca^{2+}$ events, which are stochastic in nature. We find that kiss-and-run (KR) exocytosis, which is mediated through *Syt4* and triggered by high amplitude and fast $Ca^{2+}$ events, shows much less variability when compared to full-fusions (FF) through *Syt7* (Fig 3A). This was also evident from the temporal profile of KR releases which displays faster dynamics compared to FF exocytosis (Fig 3B and 3C). The population of docked vesicles declines shortly after the stimulus and gets replenished to initial levels in a timespan of about 15–20 seconds (Fig 3D). In contrast, the population of mobile vesicles increases and remains high even after the stimulus is removed (Fig 3E). Apart from vesicle replenishment rates, the differences between KR and FF exocytosis are also evident in the dynamics of endocytosis and reacidification of docked and mobile vesicles (Fig 3D and 3E). To validate the model, we compared the temporal profile of the simulated gliotransmitter release with published experimental data on gliotransmission from several studies for similar stimulus protocols [5,61,81,82]. To make this comparison, we pooled data from 400 independent simulation trials which corresponds to the number of microdomains in the experimental data (S2 Appendix). Evidently, the temporal distribution of $Ca^{2+}$-mediated release events evoked by DHPG stimulation is in good agreement with the experimental data (Fig 3F). It is

also interesting to note that the temporal profile of gliotransmission is remarkably similar across studies that used different agonists and concentrations, suggesting that all these stimuli produced saturating responses and resulted in a complete depletion of the vesicles. (S2 Fig).

## Modeling Aβ conditions in astrocytes

We next examine signaling by astrocytes in the presence of Aβ induced molecular alterations. It has been shown that the application of Aβ enhances mGluR5 activation when stimulated by DHPG to result in roughly 50% reduction in the half-activation concentration and a near doubling of the maximal response in the dose-response curve, thereby suggesting an increase in mGluR5 expression [31,83]. However, the functional consequences of enhanced $Ca^{2+}$ signaling through this route is not clear. We incorporated these experimentally observed changes in mGluR signaling by adjusting the parameters of the mGluR model (S1 Table) that appropriately shifted the dose-response curve for maximal $IP_3$ production. Our estimates of dissociation constant ($k_d$) and peak $IP_3$ response clearly capture the halving and doubling of these quantities, respectively, as reported by previous experimental studies (Fig 4A). Independently, Aβ is also seen to inhibit PMCA function by direct binding to the calmodulin binding site [84]. Accordingly, we adjusted the backward reaction rate of the PMCA model (S1 Table) to shift the dose-response curve rightward (Fig 4B). This led to an almost 50% reduction in the $Ca^{2+}$ binding affinity of the PMCA, which is in good agreement with in vitro findings [84]. An interesting corollary to this Aβ-induced PMCA alteration is the accompanying shift in the $Ca^{2+}$ concentration that corresponds to zero net $Ca^{2+}$ flux across the PMCA. As a result of which, both resting cytosol and ER $Ca^{2+}$ concentrations are higher in astrocytic compartments with impaired PMCA activity (Fig 4C). We, therefore, propose that Aβ-induced alterations in PMCA functionality is the underlying mechanism for the observed shift in intracellular resting $Ca^{2+}$ levels in astrocytes in AD animal models [27].

We next examine the consequences of these two Aβ conditions on astrocytic $Ca^{2+}$ signaling activated by the mGluR agonist, DHPG, at different concentrations. It is clear from the heat maps (representative traces of independent simulation trials) that in the presence of DHPG (100μM, 10s), $Ca^{2+}$ signaling is considerably elevated in all Aβ conditions (Fig 4D). Quantification of three important metrics of astrocytic $Ca^{2+}$ signaling that affect gliotransmission, namely peak, event rate, and % $Ca^{2+}$ response (defined in methods), also indicated an overall increase (leftward and upward shifts in the dose-response curves) in $Ca^{2+}$ influx (Fig 4E–4G), and is in good agreement with previous experimental results, both in vivo and in vitro [85,86]. Finally, it is also evident that the presence of PMCA alteration specifically led to an upward shift in all the dose-response curves due to an increase in spontaneous $Ca^{2+}$ activity (Fig 4E–4G). We, therefore propose Aβ-induced alteration in PMCA functionality as a molecular mechanism for the experimentally reported increase in spontaneous $Ca^{2+}$ activity in astrocytes treated with Aβ [87].

## Aβ modifies astrocytic $Ca^{2+}$ signaling, gliotransmission, and the temporal relationship between them

Given the elevated $Ca^{2+}$ activity in astrocytes associated with Aβ accumulation [27,76] we investigate the $Ca^{2+}$ response to a range of stimuli that incorporates physiological concentration and decay profile of glutamate in the extracellular space (S1 Fig). Not surprisingly, $Ca^{2+}$ signaling is enhanced in astrocytic compartments with Aβ conditions. This is described in the heat maps of $Ca^{2+}$ events evoked by glutamate stimulation at 10Hz (Fig 5A). The analysis reveals interesting differences in $Ca^{2+}$ signaling that depend on the provenance of the Aβ condition. Firstly, we show that peak amplitudes of $Ca^{2+}$ events are exclusively higher in the

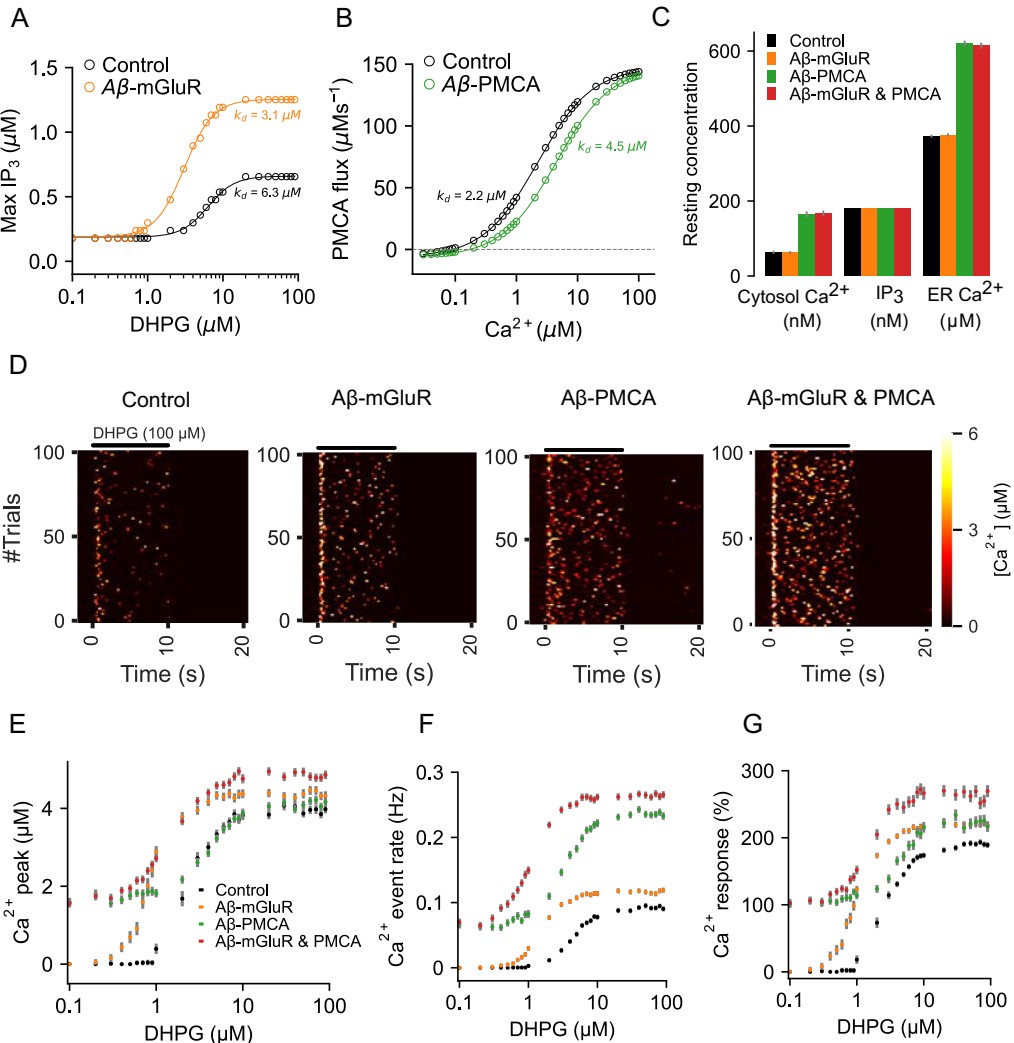

**Fig 4. Characterization of the astrocytic compartmental models with Aβ-induced molecular alterations.** (A) Peak amplitudes of IP$_3$ generated by DHPG stimulation from control and an astrocytic compartment with altered mGluR function are fitted to Hill functions for estimating dissociation constants ($k_d$). (B) Steady-state PMCA fluxes from control and Aβ-PMCA conditions fitted to Hill functions. (C) Resting cytosolic Ca$^{2+}$, IP$_3$, and ER Ca$^{2+}$ levels in control and Aβ-conditions. (D) Representative heat maps of Ca$^{2+}$ signaling from control and Aβ conditions stimulated with DHPG. (E) Peak amplitude of Ca$^{2+}$ event evoked with different DHPG concentrations. (F) Rate of Ca$^{2+}$ events evoked by different DHPG concentrations. (G) Percentage Ca$^{2+}$ response evoked by DHPG stimulation is higher the presence of Aβ conditions. Error bars indicate the standard error of the mean computed from 6 independent sets of simulation trials.

presence of the Aβ-mGluR condition. In comparison, parameters that control temporal features of Ca$^{2+}$ signaling such as rise time, FWHM, and decay time are specifically affected by the Aβ-PMCA condition and are surprisingly insensitive to the frequency of glutamate stimulation (Fig 5D–5F). Lastly, both Aβ conditions lead to an increase in the rate of Ca$^{2+}$ events (Fig 5C). In general, the model predicts several alterations in astrocytic microdomain Ca$^{2+}$ signaling in the presence of Aβ conditions that is also highly dependent on the target molecular mechanism.

We then ask how the changes in Ca$^{2+}$ signaling because of Aβ impact vesicular release by astrocytes. The raster plot of release events activated by a 10 Hz stimulus indicates that both

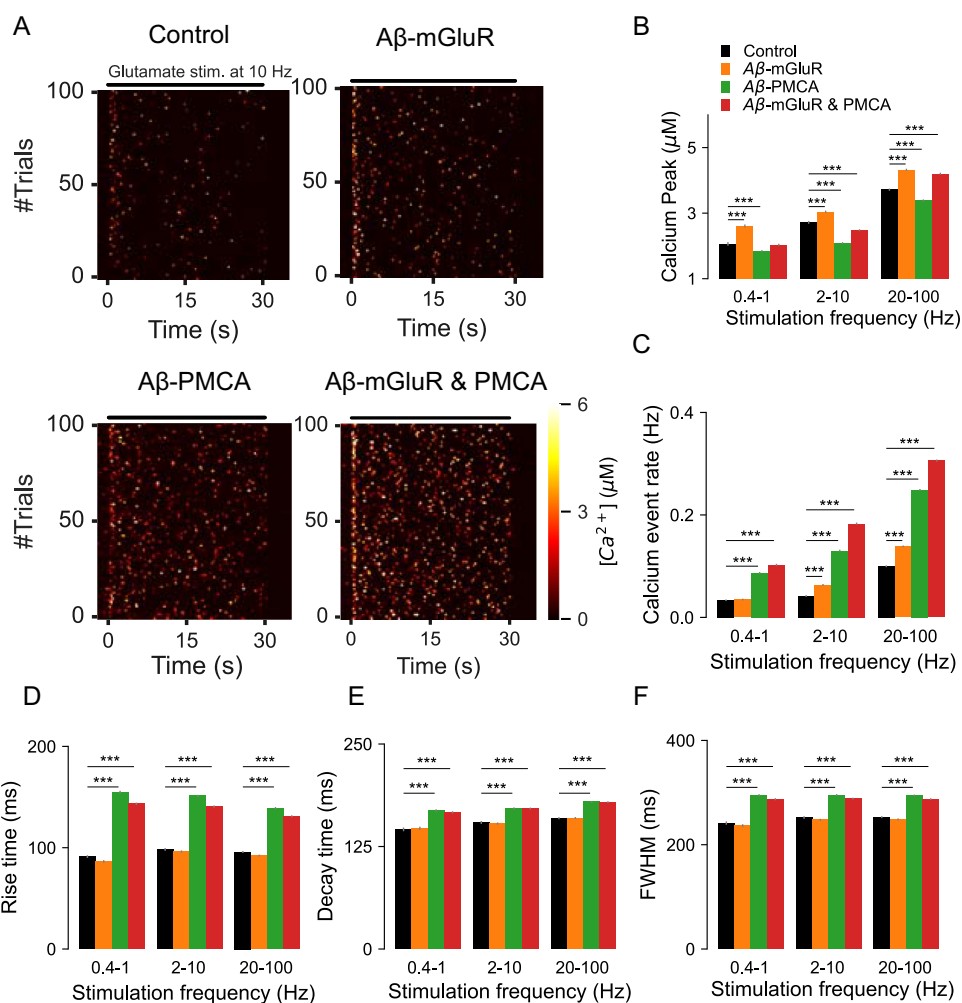

**Fig 5. Quantification of the dynamics and kinetics of Ca$^{2+}$ events from astrocytic compartments stimulated with glutamate vesicles.** (A) Representative heat maps highlight the differences in Ca$^{2+}$ dynamics in the presence of Aβ conditions. (B) Peak Ca$^{2+}$ amplitude averaged across different ranges of stimulation frequencies. (C) Glutamate-evoked Ca$^{2+}$ events are more frequent in astrocytes with Aβ conditions when compared to control. (D) The presence of Aβ-PMCA condition increases the (E) rise time, (F) decay time and (F) Full width at half maximum of astrocytic Ca$^{2+}$ events. Error bars indicate the standard error of the mean from 6 independent datasets.

kiss-and-run and full-fusion are elevated by Aβ conditions (Fig 6A). Full-fusion exocytosis is substantially elevated by the Aβ-PMCA condition (Fig 6B). In contrast, at higher stimulation rates, the Aβ-PMCA condition, but not the Aβ-mGluR condition, decreases kiss-and-run exocytosis (Fig 6C). The distinct effects of the two Aβ conditions on vesicular exocytosis are also captured in the bar graphs for a wide range of glutamate stimulation rates.

Synchronous Ca$^{2+}$ activity between astrocytic subpopulations is observed in animal models of AD [29]. Locally synchronized astrocytic Ca$^{2+}$ rises are also thought to play an important role in the manifestation of seizure-like events [88]. Therefore, we investigated the impact of Aβ conditions on the timing of Ca$^{2+}$ and gliotransmitter release events across independent simulation trials by computing temporal synchrony between events at different interevent intervals (see Methods). In the absence of Aβ conditions, Ca$^{2+}$ events are not synchronized when the frequency of stimulation is low but gets moderately synchronized at higher stimulation rates (Fig 7A). However, this pattern is altered in the presence of Aβ conditions.

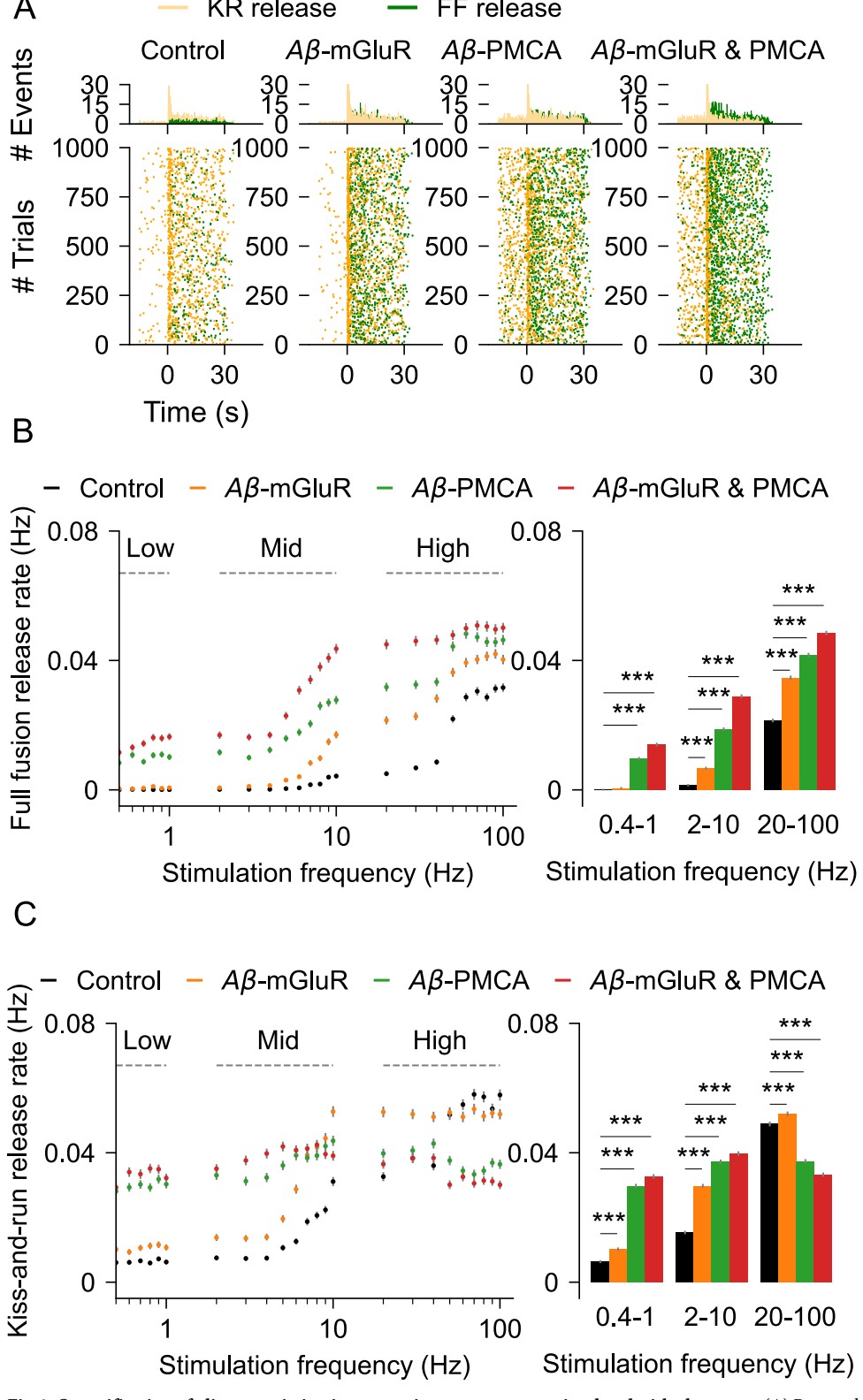

**Fig 6. Quantification of gliotransmission in astrocytic compartments stimulated with glutamate.** (A) Raster plots of kiss-and-run and full-fusion releases in control astrocyte and in the presence of Aβ conditions. *Top*: temporal histograms. (B) Rate of full fusions for different glutamate stimulation rates. *Right*: release rates for the three stimulus groups (C) Rate of kiss-and-run exocytosis. Right: release rates pooled into three simulation groups. Error bars indicate the standard error of the mean computed from 6 independent datasets.

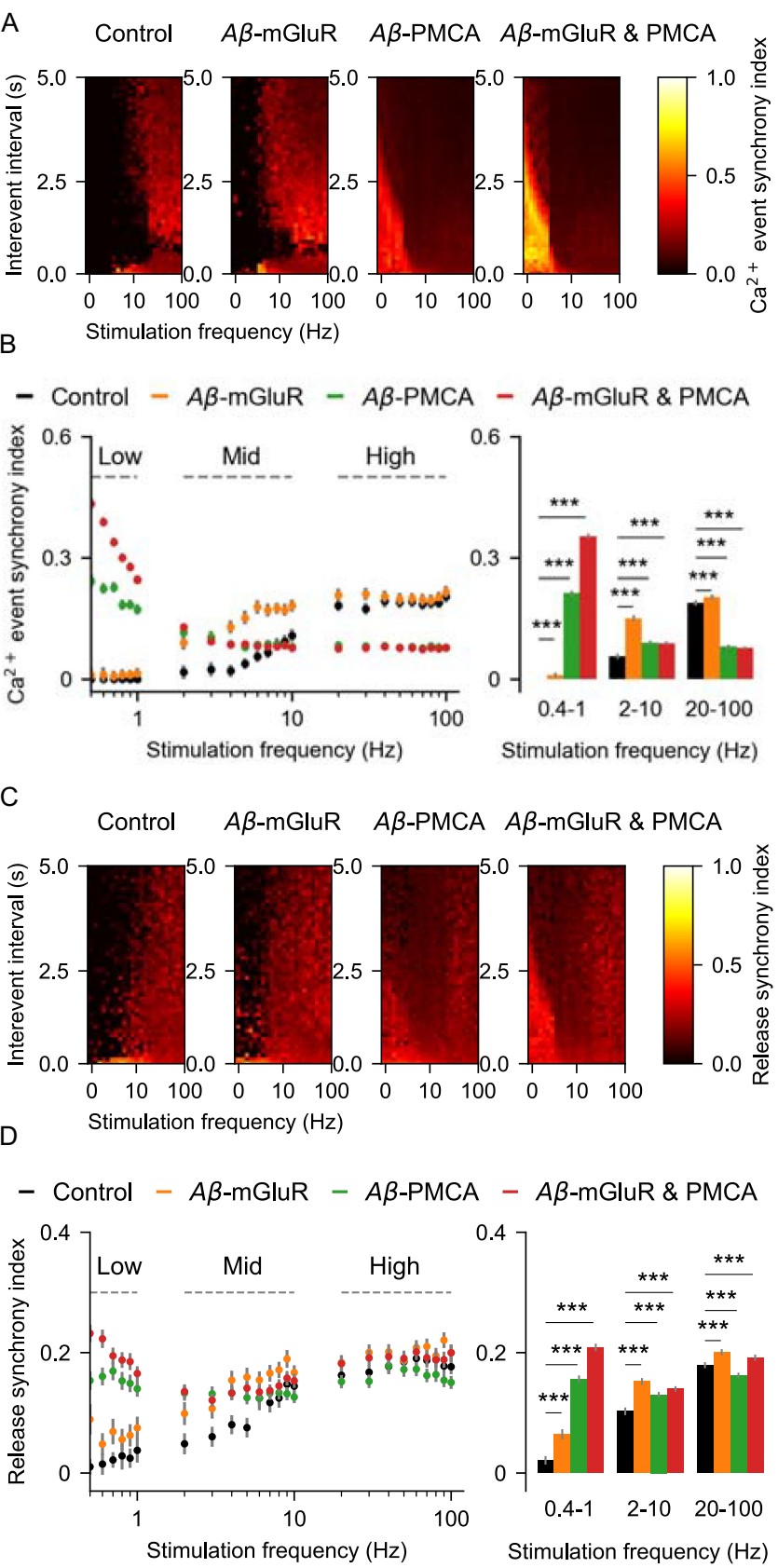

**Fig 7. Synchrony of Ca$^{2+}$ and gliotransmitter release events from astrocytic compartments stimulated with glutamate.** (A) Heat maps of Ca$^{2+}$ event synchrony at different inter-event intervals and stimulation frequencies in the presence and absence of Aβ conditions. (B) Ca$^{2+}$ synchrony averaged across event intervals at different glutamate stimulation rates is different between control and Aβ conditions. *Right*: at low stimulation regime, synchrony is high with the Aβ-PMCA condition, whereas at high stimulation rates, synchrony is more in the Aβ-mGluR condition (C) Heat maps of release event synchrony at different inter-event intervals and stimulation frequencies in the presence and absence of Aβ conditions. (D) synchrony of release averaged across event intervals at different glutamate stimulation rates is different between control and Aβ conditions. *Right*: bar graph indicates the increase in release synchrony at different stimulation regimes in all the Aβ conditions. Error bars indicate the standard error of the mean computed by bootstrapping 100 times with a sample size of 1000.

Particularly, we show that the presence of the Aβ-PMCA condition is associated with highly synchronous Ca$^{2+}$ events at low frequencies of glutamate stimulation but promotes desynchrony at higher rates (Fig 7B). This shift in synchronicity by Aβ conditions under different stimulation rates is also evident in the bar plots that show synchrony averaged across three different stimulation regimes. While global Ca$^{2+}$ activity between astrocytes becomes synchronous in the presence of Aβ accumulation [29], the molecular mechanism is not clear. Our results indicate Aβ-mediated changes in PMCA functionality as a potential mechanism.

The synchrony of vesicular release events is also affected by Aβ conditions (Fig 7C). Namely, peak synchrony of vesicular release is shifted leftward towards low the frequency regime in the presence of the Aβ-PMCA condition (Fig 7D).

The temporal correlation between Ca$^{2+}$ and gliotransmitter events is critical for astrocytic modulation of synaptic activity at short timescales [23,89,90]. Our simulations clearly demonstrate that Aβ is a modulator of both Ca$^{2+}$ signaling and gliotransmission (Figs 5 and 6). How does the precise temporal correlation that was reported by experimental studies and reproduced by our model affected by Aβ conditions [5]? We investigate this by computing the correlation between Ca$^{2+}$ and release events across independent simulation trials. As expected, cross-correlation is high in the control condition at low to moderate levels of stimulation rates (Fig 8A). In striking contrast, cross-correlation is low in the presence of Aβ-PMCA condition, and it is at the lowest in the presence of both PMCA and mGluR conditions. This is surprising because independently both the Aβ conditions increase Ca$^{2+}$ events and gliotransmitter release events.

Apart from Ca$^{2+}$ event amplitudes, gliotransmission is also dependent on the availability of vesicles. We, therefore, examined changes in the vesicle resource as the root cause of decrease in cross-correlation. We see that the presence of Aβ conditions lowers the population of docked vesicles at high stimulation frequencies (Fig 8B) and mimics the trend of decrease in cross-correlation (Fig 8A). Thus, we show that the presence of Aβ conditions, despite causing an increase in both Ca$^{2+}$ and release events, leads to the loss in the temporal relationship due to the rapid depletion of docked vesicles. Further quantification indicates the exponential relationship between peak cross-correlation and docked vesicles that is most catastrophic for the Aβ-PMCA condition (Fig 8C). In line with our speculation on the mechanism for the loss of the temporal correlation between Ca$^{2+}$ and gliotransmitter release events, we have also shown that kiss-and-run exocytosis of docked vesicles is heavily impacted by the Aβ-PMCA condition (Fig 6C).

## Discussion

The model of a single compartment of an astrocyte described here accurately captures the temporal characteristics of Ca$^{2+}$ signaling and the resulting gliotransmission as observed in experimental studies [5,11]. To the best of our knowledge, it is the first model to include a biophysical description of two calcium sensors (*Syt4* and *Syt7*), two release modes (full-fusion and kiss-and-run), and vesicle recycling (Figs 1–3). Aβ modulates Ca$^{2+}$ signaling in astrocytes both in vivo [29] and in vitro [31,34,83]. Aβ is also a well-known regulator of presynaptic

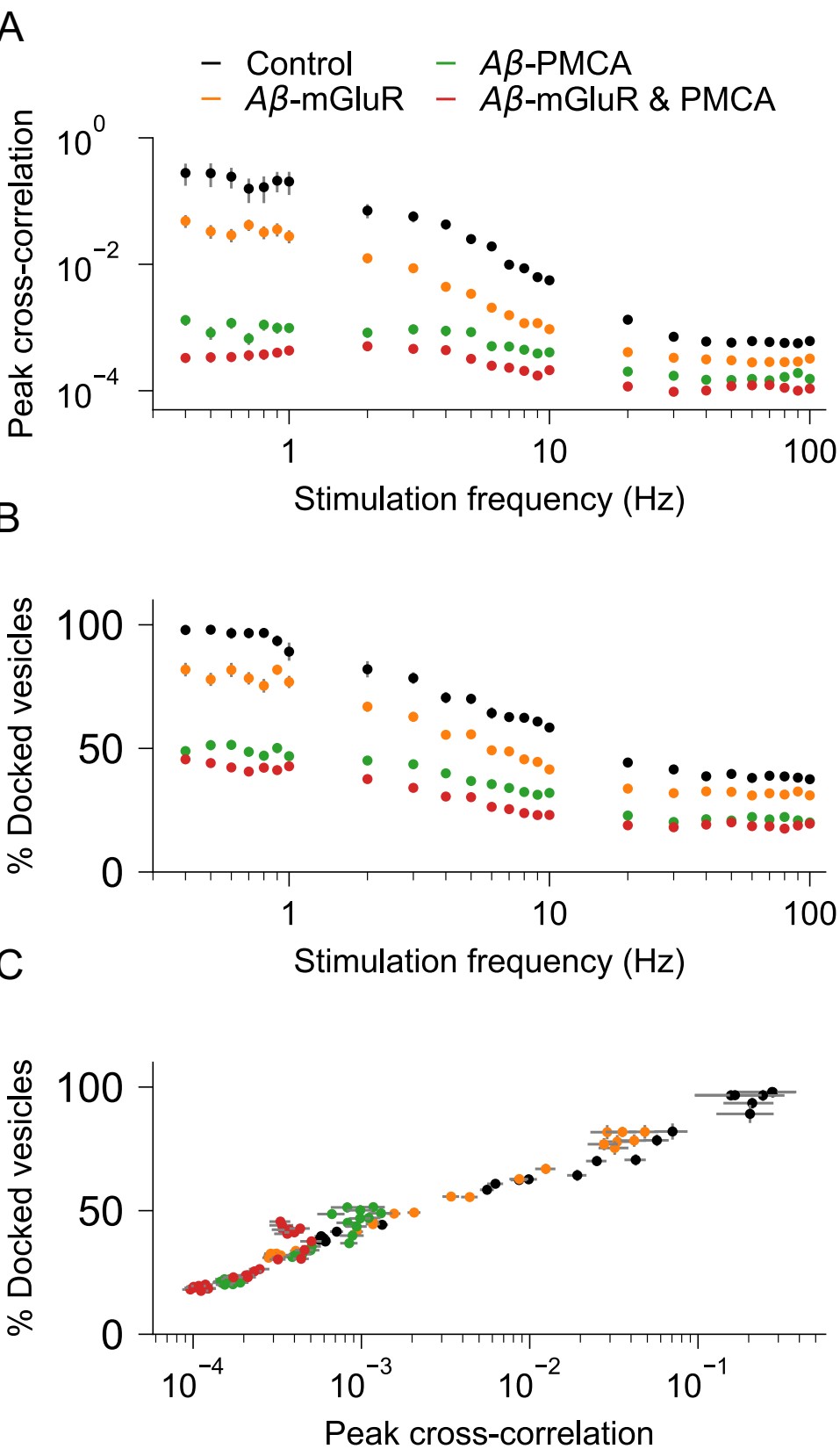

**Fig 8. Cross-correlation between individual Ca$^{2+}$ and gliotransmitter release events are disrupted in astrocytic compartments with Aβ pathology.** (A) Peak cross-correlations between Ca$^{2+}$ and release events become low at higher stimulation rates in astrocytes with Aβ conditions when compared to control. (B) Higher stimulation rates decrease population of docked vesicles in astrocytes with Aβ conditions when compared to control. (C) The presence of Aβ conditions shifts the relationship between peak cross-correlation and docked vesicle population.

release probability [91]. However, mechanistic links between Aβ, astrocytic Ca$^{2+}$ signaling, and gliotransmission are not known. The model reveals the distinct effect of two calcium signaling mechanisms, namely mGluR and PMCA, when modulated by Aβ on (1) characteristics of Ca$^{2+}$ spikes, (2) Ca$^{2+}$-mediated gliotransmission, (3) populations of docked and mobile vesicles, and (4) synchrony of Ca$^{2+}$ and release events between trials. While both mechanisms increase the rate of Ca$^{2+}$ events, changes in the mGluR signaling pathway elevate the amplitude of the Ca$^{2+}$ events, whereas alteration in PMCA function increases the duration of the Ca$^{2+}$ events (Fig 5). Further, the underlying high amplitude and fast Ca$^{2+}$ events in the Aβ-mGluR condition favor kiss-and-run exocytosis through *Syt4*. This is because the Ca$^{2+}$ affinity for *Syt4* is low compared to *Syt7*. In contrast, the long-lasting Ca$^{2+}$ events in Aβ-PMCA condition promote full-fusion releases through *Syt7* (Fig 6). The complete model with both PMCA and mGluR pathology captures the increase in synchrony of Ca$^{2+}$ events between trials (Figs 6 and 7), consistent with experimental observation [29].

The correspondence between a calcium event and a vesicle release event is a characterizing feature of gliotransmission under physiological conditions. One of the most interesting observations from the model is the breakdown of this tight temporal relationship between Ca$^{2+}$ and release events under the conditions of modified calcium signaling corresponding to the presence of Aβ (Fig 8). Our study predicts Aβ-induced depletion of docked vesicles as the possible mechanism for this disruption in signaling by astrocytes.

## Building a control model

We build on previous models of intracellular calcium signaling to seek an accurate description of synaptically-activated gliotransmission at individual perisynaptic astrocytic processes. The model includes realistic descriptions of Ca$^{2+}$ buffers and pumps, both within the cytoplasm and ER, and a stochastic activity of IP$_3$Rs that generate physiologically realistic Ca$^{2+}$ events. The model further incorporates available experimental data on (1) the number of calcium-binding sites and the calcium-binding affinities of each of the two calcium sensors, (2) the presence of two distinct vesicle populations, (3) the two distinct pathways of vesicle recycling and associated timescales (S1 Table and S1 Appendix). We have illustrated here the value of a biologically realistic model to investigate and quantify the link between Aβ-induced molecular mechanisms and their functional implications. It is widely accepted that astrocytic compartments next to synapses are equipped with several unique features such as (1) increased ER presence near docked vesicles, (2) high levels of mGluRs expression, and (3) IP$_3$Rs that together allow the generation of Ca$^{2+}$ events with high amplitudes and fast kinetics [5,6,92]. Other studies also suggest that Ca$^{2+}$ dynamics in microdomains are several orders faster when compared to global somatic Ca$^{2+}$ signals [22,60,93]. Notably, our model predictions on characterizing features of Ca$^{2+}$ events such as peak amplitude, rise time, decay time, and full width at half maximum are in line with these experimental findings [5,10].

## Predicting AD-associated changes in calcium signaling

Our model reproduces an increase in resting Ca$^{2+}$ levels and a rise in spontaneous and synchronous high amplitude Ca$^{2+}$ events, all in good agreement with experimental findings on

Aβ-induced alterations [11,29]. Additionally, our findings also suggest Aβ-induced changes in PMCA functionality as an underlying mechanism for the above effects (Fig 4). Aberrations in the activity of PMCAs, apart from elevating cytosolic $Ca^{2+}$, also sequesters $Ca^{2+}$ into the ER, thereby increasing resting ER $Ca^{2+}$ concentration. $Ca^{2+}$ responses in astrocytes have a critical role in arterial blood flow regulation [94], and separately, amyloid depositions also impair blood flow [95], however, the underlying mechanisms are not clear. An increase in intracellular $Ca^{2+}$ by Aβ-PMCA condition as described here can potentially impact cerebral blood flow and hence brain function. Our study also supports the proposed role of the Aβ-PMCA condition (and not Aβ-mGluR) in the generation of spontaneous $Ca^{2+}$ activity in astrocytes [11]. Like neurons, astrocytes can also dynamically, as per need, switch their mode of exocytosis consistently, from kiss-and-run to full-fusion in response to different stimuli [20,96]. Although the underlying mechanism is not clear, it highlights the specific function of each mode of release. We show that the Aβ-PMCA condition drastically reallocates the rates of full-fusion and kiss-and-run. Specifically, there is a shift to full-fusion mode from kiss-and-run mode, and this shift decorrelates gliotransmitter release from the underlying $Ca^{2+}$ events. The Aβ-PMCA condition generates $Ca^{2+}$ events that are long-lasting with comparatively large rise time, full-width, and decay time. This results in increased *Syt7* activation as the reaction kinetics of this sensor are slower than *Syt4*. These results, apart from highlighting the importance of $Ca^{2+}$ dynamics in shaping gliotransmission, also shed light on the distinct ways by which the two Aβ conditions affect both $Ca^{2+}$ signaling and gliotransmitter release.

## Other insights from the model

While gliotransmitter releases have been observed at highly ramified astrocytic processes, they are still experimentally difficult to study. Despite this challenge, several studies have provided excellent data on the characteristics of $Ca^{2+}$ and release events from single microdomains. It has been shown that at single processes, there is a remarkable temporal correlation between individual $Ca^{2+}$ and release events [5]. However, it is important to note that these processes have a ninefold lower capacity of docked vesicles when compared to their presynaptic counterparts–the axon terminals [89]. This suggests that vesicles in astrocyte processes are extremely valuable 'commodities'. While several studies have reported on the increase in $Ca^{2+}$ signaling and, to a lesser extent, on the gliotransmission because of Aβ accumulation, the implications of having a low population of docked vesicles have not been considered either for normal physiology or disease conditions. We show that the temporal correlation between vesicle releases and calcium events is heavily dependent on the population of docked vesicles, and their non-linear relationship is altered in the presence of a disease condition–the Aβ pathology. Another aspect that is less understood is the presence of mobile vesicles in astrocytes [97,98] and their temporal dynamics. We show that the population of mobile vesicles increases shortly after the stimulus and takes tens of seconds to return to baseline levels. It has also been shown that the mobility of astrocytic vesicles increases with changes in astrocytic $Ca^{2+}$ levels; our simulation corroborates this observation [99].

## Justification for modeling framework

Multiple evidence supports our model for gliotransmission with two synaptotagmins (*Syt4* and *Syt7*), that differ in both their reaction kinetics and $Ca^{2+}$ affinities (details are in S3 Appendix) [15,68]. As suggested by experiments, the model includes separate pools of docked and mobile vesicles that are distinctly targeted by *Syt4* and *Syt7*, respectively [5,19,73]. Our choice of model parameters that control the timescales of exocytosis, endocytosis, and reacidification rates are based on experimental measurements of kiss-and-run and full-fusion exocytosis

[5,18]. The synaptotagmin *Syt4* is exclusively activated by fast and high amplitude $Ca^{2+}$ signals and primarily triggers fast temporally correlated release (synchronous release) of docked vesicles via kiss-and-run mode. This is in line with the suggestion that the ER membrane is juxtaposed onto docked vesicles to allow kiss-and-run release by fast and high amplitude $Ca^{2+}$ events [5]. On the other hand, *Syt7*, which has high $Ca^{2+}$ affinity but slow kinetics, is activated by low amplitude and slow $Ca^{2+}$ events to cause release events that are temporally decorrelated with calcium events (asynchronous release). Also, unlike other synaptotagmins that are attached to the vesicle itself, *Syt7* is diffusely expressed on the plasma membrane [100]. These features allow *Syt7* to target mobile vesicles, which are not only reported in astrocytes but are also known to be transported in an activity-dependent manner [97,99]. A considerable portion of these vesicles become immobile once they reach the distal end of processes, suggesting vesicle docking [73,101]. Our model, in line with these findings, implements fast synchronous release of docked vesicles and slow, the asynchronous release of mobile vesicles through *Syt4* and *Syt7*, respectively [20,102].

Glutamate, the major excitatory neurotransmitter in the brain, has also been heavily implicated in the neuropathology of AD. While the bulk of the studies examining the synaptic component of Aβ pathology is focused on neurons, there is growing evidence that shows astrocytes contribute to the excitotoxicity observed in AD brains [103]. Astrocytes are primarily responsible for the clearance and metabolism of synaptically-released glutamate, and interestingly enough, both glutamate and glutamine levels are altered in AD brains [104]. Since astrocytes outnumber neurons in most brain regions, an increase in gliotransmission through the Aβ toxicity can cause major imbalances in glutamate levels and accelerate synaptic loss through excitotoxicity [105]. Increases in extracellular glutamate via gliotransmission are also implicated in the generation of $Ca^{2+}$ waves owing to the tightly interconnected network architecture [106]. Modified gliotransmitter release, as predicted by our model, can alter the spatial extent of the $Ca^{2+}$ waves. Interestingly $Ca^{2+}$ waves are known to specifically originate from sites near Aβ plaques [29]. Since astrocytes are electrically silent, the tight temporal correlation of $Ca^{2+}$ events with the gliotransmitter release is critical for astrocytic feedback [107]. The prediction from our study that describes disruption of the temporal correlation between $Ca^{2+}$ and gliotransmission by Aβ mechanisms can have severe downstream consequences on astrocyte-neuron signaling. This also strengthens the idea that modified signaling by astrocytes contributes to AD pathology. It has been suggested that one of the roles of asynchronous release in neurons is to desynchronize the activity of downstream neurons [108]. Considering that astrocytic compartments near synapses modulate neural firing through gliotransmitters, the observed loss in a temporal relationship can alter synaptic signaling and negatively impact information transfer across synapses in AD.

In summary, we developed a biophysical model that accurately describes $Ca^{2+}$ signaling and concomitant gliotransmitter release in a single astrocytic compartment. We show that Aβ, by acting on the astrocytic $Ca^{2+}$ signaling and gliotransmission machinery, derails astrocyte to neuron communication. Future experiments are necessary to test whether this is one of the mechanisms through which Aβ deposits impair cognitive function in AD patients.

## Supporting information

**S1 Appendix. Details of the model.**
(DOCX)

**S2 Appendix. Estimation of the number of astrocytic compartments in the experimental data.**
(DOCX)

**S3 Appendix. Calcium sensors in the gliotransmitter release model.**
(DOCX)

**S1 Fig. Time courses of glutamate, IP$_3$ and Ca$^{2+}$ concentrations.**
(DOCX)

**S2 Fig. Temporal profiles of vesicle release from astrocytic domains in response to various stimuli.**
(DOCX)

**S3 Fig. Kinetic schemes of astrocytic calcium sensors (synaptotagmins).**
(DOCX)

**S1 Table. Model parameters.**
(DOCX)

## Author Contributions

**Conceptualization:** Anup Gopalakrishna Pillai, Suhita Nadkarni.

**Data curation:** Anup Gopalakrishna Pillai.

**Formal analysis:** Anup Gopalakrishna Pillai.

**Funding acquisition:** Anup Gopalakrishna Pillai, Suhita Nadkarni.

**Investigation:** Anup Gopalakrishna Pillai.

**Methodology:** Anup Gopalakrishna Pillai.

**Project administration:** Suhita Nadkarni.

**Resources:** Suhita Nadkarni.

**Software:** Anup Gopalakrishna Pillai.

**Supervision:** Suhita Nadkarni.

**Validation:** Anup Gopalakrishna Pillai.

**Visualization:** Anup Gopalakrishna Pillai.

**Writing – original draft:** Anup Gopalakrishna Pillai.

**Writing – review & editing:** Anup Gopalakrishna Pillai, Suhita Nadkarni.

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
