## [Decision Letter · Decision Letter 0]

25 Aug 2020

Dear Dr Nadkarni,

Thank you very much for submitting your manuscript "Amyloid pathology disrupts gliotransmitter release in astrocytes" for consideration at PLOS Computational Biology.

As with all papers reviewed by the journal, your manuscript was reviewed by members of the editorial board and by several independent reviewers. In light of the reviews (below this email), we would like to invite the resubmission of a significantly-revised version that takes into account the reviewers' comments. This revised version should take into account all the issues pointed by the 4 reviewers, including those by reviewer 1 in their detailed comments provided as an attached pdf file (regarding the statement of the research goal, the use of references or the quality of English).   

Moreover, since your article was submitted within the Reproducibility Modelling Pilot scheme, it is crucial that your code be made available to the reviewers in order for them to check the reproducibility of your results.  

We cannot make any decision about publication until we have seen the revised manuscript and your response to the reviewers' comments. Your revised manuscript is also likely to be sent to reviewers for further evaluation.

Sincerely,

Hugues Berry

Associate Editor

PLOS Computational Biology

Daniele Marinazzo

Deputy Editor

PLOS Computational Biology

Reviewer's Responses to Questions

**Comments to the Authors:**

Reviewer #1: Pillai and Nadkarni introduce a novel model of astrocyte-synapse interactions that revolves around the biophysical description of calcium-dependent glutamate release from astrocytes. In particular, they distinguish between different mechanisms of glutamate exocytosis – kiss-and-run vs. full-fusion – by considering the Ca2+-dependent dynamics of different synaptotagmins. Then, they proceed to mimic amyloid-like pathology changing values to model parameters, and documenting, through numerical simulations, how amyloid-beta could affect astrocytic physiology, and astrocytic glutamate release in particular. The study is original and of interest to PLoS Computational Biology, with the potential to reach out to a broad audience, comprising both experimentalists and computationalists. Nonetheless, there is ample space for improvement of the manuscript in its present version. There are at least three main issues with the present paper. First, the research question is not clear, and the way the material is exposed is not exhaustive and somehow lousy. Second, there is a critical issue with the references cited by the authors, which often do not support the authors’ claims. Third, the writing style is poor, making the reading of the manuscript hard for several broken sentences, ambiguous terminology, and logical flow that is not consistent. For these reasons, I am advising for major revisions, which I am detailing in the attachment.

Reviewer #2: Pillai and Nadkarni develop a computational model to link Ca2+ signaling in the processes of astrocytes with the gliotransmitter release by these cells. The model has two main components: (1) Ca2+ dynamics in astrocytes and (2) gliotransmitter release. A rise in Ca2+ concentration in the processes of astrocytes leads to transmitter release through a fast release pathway (termed kiss-and-run mechanism) and slow release pathway. The model reproduces key observations about Ca2+ signals and distribution of transmitter release events. The model is then used to explore the effect of changes in two Ca2+ pathways observed on in Alzheimer’s disease on Ca2+ dynamics. These include reduced plasma membrane Ca2+ ATPase (PMCA) and gain-of-function enhancement of metabotropic glutamate receptors (mGluRs). The model provides some useful insights into how these two pathways affect Ca2+ signaling and gliotransmitter release in Alzheimer’s disease-affected astrocytes.

In my view the main contribution of this study is the development of gliotranmistter release model. Overall, I find the study very interesting as it is addressing an important topic. However, there are some key issues that needs to be addressed before accepting the paper for publication.

Page 5: In the Introduction section, the authors state “While extensive experimental data and computational studies exist on somatic Ca2+ dynamics in astrocytes [26,27], a quantitative formulation of Ca2+ dynamics at a single astrocytic process and its relationship to Ca2+ mediated kiss-and-run and full fusion exocytosis has not been described”. I find this statement a little tricky. The second part of the statement is true, i.e. “a quantitative formulation of Ca2+ dynamics at a single astrocytic process and its relationship to Ca2+ mediated kiss-and-run and full fusion exocytosis has not been described”. However, the first part of the statement makes it sounds like the authors are doing something completely different with the Ca2+ dynamics to model Ca2+ signaling in the process. The fact is, both the “somatic Ca2+ dynamics in astrocytes” models developed before and the model presented in this paper are point models and none pay attention to the spatial structure. Thus the notion that their Ca2+ model is a radical departure from the previous models is technically not correct.

Page 6: Before equation 1 “….therefore on included in the present model” should be “….therefore not included in the present model”

Page 8-10: There is a disconnect between the description of Ca2+ model and gliotransmitter model. A clear description is needed explaining how the kinetic scheme in Figure 2B is related to the release and recycling scheme. For example, how do the “Docked vesicles” or “Release vesicles” relate to the scheme “S0 <->S1 <->S2”? Same applies to Syt7 scheme.

Similarly, what does the statement “We characterized the release machinery by clamping intracellular Ca2+ levels and measuring peak release (R) rate, which was fitted to a Hill equation (4) with Vmax, Kd and coeff as free parameters” mean for the model? And how is equation (4) connected to the gliotransmitter model?

How were the rate equations and parameters in the gliotransmitter model optimized? Were they optimized using trial and error method using the full model (i.e. Ca2+ and gliotransmitter dynamics together)? Or some kind of automated optimization method employing modular approach was used?

Page 10-12: I believe the stochasticity in the synchrony between Ca2+ events and release come from the stochastic modeling of gliotranmission. However, this hasn’t been explicitly mentioned in the paper.

Page 14, Figure 1: More details about the implementation of DHPG in the model are needed. How is DHPG related to glutamate or mGluR?

Was glutamate raised to a fixed value for 2 sec or was raised for 2 sec and allow to decay according to first order kinetics? Was the rise a single event or multiple events applied periodically over 2 seconds?

Page 14, 15, Figure 2B: The statement “Syt4 has a single low-affinity domain (C2B) that binds 2 Ca2+ ions with fast forward rates” is consistent with the kinetic scheme in Figure 2B. However, it is not clear if the mechanism described by the statement “It is also established that Syt4 with a single Ca2+ binding domain can only promote kiss-and-run exocytosis [16]” has been implemented in the model or not. If it is, how? Do states S1 and S2 in the Syt4 scheme both lead to vesicles release? Are the release probabilities in both S1 and S2 states the same or different?

Overall, I believe it would make it easier to follow the model if all the rate equations for the gloitransmitter model are listed in a new table.

Page 18 and Figure 5: What was the amplitude and life time (duration for which glutamate was raised) of each glutamate release (spillover)? I believe the amplitude of each individual glutamate release (spillover) event was ~200 uM and was modeled as first order rate equation with a 6.25 ms rise time and 10 ms decay time throughout the paper? If so, this should be explicitly stated on page 9.

Page 20: The reference (kuchibhotla2008) needs to be fixed.

Line 1, page 24: “ER Ca2+ levels (400 mM)” should be “ER Ca2+ levels (400 uM)”?

Reviewer #3: Reproducibility report has been uploaded as an attachment.

Reviewer #4: This work is a model study of cellular mechanisms associated with Alzheimer's disease.

Elucidation of the causes of excessive accumulation of beta-amyloid and the mechanisms of cleansing brain tissue is currently a hot topic in brain physiology . At the same time, another important part in the area is to establish the cellular mechanisms that accompany the accumulation of beta-amyloids. On this basis, we can conclude that the problem solved in this work is of interest as a small puzzle piece that contributes to a general understanding of the problems associated with Alzheimer's disease. It is valuable for the specialists in the field, while might be less understandable for the general public.

The manuscript contains a very detailed, even meticulous, description of the modeled cellular pathways and the rationale for the choice of parameters. This allows me to agree with the authors' assertion that they "presented a detailed biophysical model" of the processes under consideration. According to my knowledge, this is the first model to quantitatively predict the release of gliotransmitter in response to beta-amyloid accumulation.

From the results obtained, I would single out two. The first is the answer to the question of what shifts in calcium dynamics are caused by the changes characteristic of the accumulation of beta amyloid. On a qualitative level, this answer could be obtained using a simpler model.

The second result is a aberration of the causal relationship (worsening correlation) by bursts of calcium and the release of gliotransmitter.

In line with the above, my assessment of the work presented is positive. I have no serious comments, and I think that the work

can be accepted for publication.

Nevertheless, I recommend that the authors consider the following comment: the discussion of the discussion is quite lengthy, and in the absence of short Conclusions is difficult to read. I would suggest either split the discussion into subsections with subtitles, or add conclusions with a really short summary.

**Have all data underlying the figures and results presented in the manuscript been provided?**

Reviewer #1: **No: **There are at least three instances where relevant data were not shown.

Reviewer #2: Yes

Reviewer #3: None

Reviewer #4: Yes

PLOS authors have the option to publish the peer review history of their article (what does this mean?). If published, this will include your full peer review and any attached files.

Reviewer #1: No

Reviewer #2: **Yes: **Ghanim Ullah

Reviewer #3: **Yes: **Anand K. Rampadarath

Reviewer #4: No
---

## [Decision Letter · Decision Letter 1]

19 Nov 2021

Dear Dr Nadkarni,

Thank you very much for submitting your manuscript "Amyloid pathology disrupts gliotransmitter release in astrocytes" for consideration at PLOS Computational Biology.

As with all papers reviewed by the journal, your manuscript was reviewed by members of the editorial board and by several independent reviewers. In light of the reviews (below this email), we would like to invite the resubmission of a significantly-revised version that takes into account the reviewers' comments.

As you will see in these reviews, Reviewer#1 still points a number of issues with your manuscript. We feel that the sarcastic and sometimes aggressive tone of this review is counterproductive and does not contribute to create the serenity that one expects in a review process. This being said, we must admit that we agree with the substance (though not the form) of several of the points raised. In particular, the absence of line or page numbers or of change tracking methods in your revised manuscript makes the reviewer's task quite painful (and change tracking was explicitly required in the decision email). We also agree with the need to tone down some of the affirmations made in the text, in order to not present as consolidated facts in the literature what is actually only current possible interpretations of experimental results (for instance, the link between gliotransmission and amyloid beta, or the hypothesis that astrocytes are critical for information processing at synapses etc). Please take also into account the reviewer's comments about the title or the pertinence of the citations and make sure that your mathematical notations are consistent throughout the text. It is important that all the reviewer's comments are addressed in your revision. You are not expected to agree and conform with every and all of these comments, but you have to address them all.

We cannot make any decision about publication until we have seen the revised manuscript and your response to the reviewers' comments. Your revised manuscript is also likely to be sent to reviewers for further evaluation.

[1] A letter containing a detailed list of your point-by-point responses to the review comments and a description of the changes you have made in the manuscript. Please note while forming your response, if your article is accepted, you may have the opportunity to make the peer review history publicly available. The record will include editor decision letters (with reviews) and your responses to reviewer comments. If eligible, we will contact you to opt in or out.

Sincerely,

Hugues Berry

Associate Editor

PLOS Computational Biology

Daniele Marinazzo

Deputy Editor

PLOS Computational Biology

Reviewer's Responses to Questions

**Comments to the Authors:**

Reviewer #1: I appreciate the effort put by the authors in improving their manuscript with respect to the original submission. However, while I found improved figures and Result's exposition, I am dismayed by the pervasive sense of carelessness still emerging from the writing style. Again, Citations appear to be often made superficially and lack a clear rationale with the cited work. The logical flow in crucial sections like Introduction and Discussion remains fragmented. This is nerving. Very often, when I read your new manuscript, I came to wonder if you were aware of what you mean in English or not since, on several occasions, your sentences do not make any sense, and your reasoning is flawed. We are reaching a stage in the computational glioscience literature where we are in charge of keeping the field rigorous and prioritizing quality and scientific rigor over bombastic statements and lousy material. While your editing would go towards the former, what still emerges from your manuscript is the latter. I am once more asking for major editings. The hope is that, since the first round took you about a year to be addressed with debatable results in terms of quality, this second occasion could instead be managed more effectively with the maximal outcome: i.e., we could finally go into publication. If not, frankly, my level of frustration with this work is reaching saturation.

On a side note, on this round, not only do you not provide a manuscript with numbered lines making exact referencing to parts of the text impossible, but also you do not number pages, making the whole reviewing effort an even more daunting task. Thank you for showing respect and appreciation for your reviewers' time and effort. In your response to the reviewers and me, I also appreciated how you replied to several points by literally pasting the same answer.

Reviewer #2: No more comments.

Reviewer #3: Reproducibility report has been uploaded as an attachment.

**Have the authors made all data and (if applicable) computational code underlying the findings in their manuscript fully available?**

Reviewer #1: Yes

Reviewer #2: Yes

Reviewer #3: Yes

PLOS authors have the option to publish the peer review history of their article (what does this mean?). If published, this will include your full peer review and any attached files.

Reviewer #1: No

Reviewer #2: **Yes: **Ghanim Ullah

Reviewer #3: **Yes: **Anand K. Rampadarath
---

## [Editor Report · Decision Letter 2]

28 Jun 2022

Dear Dr. Nadkarni,

We are pleased to inform you that your manuscript 'Amyloid pathology disrupts gliotransmitter release in astrocytes' has been provisionally accepted for publication in PLOS Computational Biology.

We would like to apologize again for the unprofessional, contemptuous and unacceptable tone of reviewer #1. We agree with a large part of the points you mention regarding the issue; actually this Associate Editor has decided to ban this reviewer from the list of reviewers he will send manuscripts to in the future.

Best regards,

Hugues Berry

Associate Editor

PLOS Computational Biology

Daniele Marinazzo

Deputy Editor

PLOS Computational Biology

---

## [Editor Report · Acceptance letter]

24 Jul 2022

PCOMPBIOL-D-20-01194R2 

Amyloid pathology disrupts gliotransmitter release in astrocytes

Dear Dr Nadkarni,

I am pleased to inform you that your manuscript has been formally accepted for publication in PLOS Computational Biology. Your manuscript is now with our production department and you will be notified of the publication date in due course.

With kind regards,

Olena Szabo
